# ‘We Have Guidelines, but We Can Also Be Artists’: Neurologists Discuss Prognostic Uncertainty, Cognitive Biases, and Scoring Tools

**DOI:** 10.3390/brainsci12111591

**Published:** 2022-11-21

**Authors:** Luca Tolsa, Laura Jones, Patrik Michel, Gian Domenico Borasio, Ralf J. Jox, Rachel Rutz Voumard

**Affiliations:** 1Chair of Geriatric Palliative Care, Lausanne University Hospital and University of Lausanne, 1011 Lausanne, Switzerland; 2Stroke Center, Neurology Service, Lausanne University Hospital and University of Lausanne, 1011 Lausanne, Switzerland; 3Palliative and Supportive Care Service, Lausanne University Hospital and University of Lausanne, 1011 Lausanne, Switzerland; 4Institute of Humanities in Medicine, Lausanne University Hospital and University of Lausanne, 1011 Lausanne, Switzerland

**Keywords:** severe stroke, decision making, prognostic uncertainty, cognitive biases

## Abstract

Introduction: Ischemic stroke is a leading cause of disability and mortality worldwide. As acute stroke patients often lose decision-making capacity, acute management is fraught with complicated decisions regarding life-sustaining treatment (LST). We aimed to explore (1) the perspectives and experiences of clinicians regarding the use of predictive scores for LST decision making in severe acute stroke, and (2) clinicians’ awareness of their own cognitive biases in this context. Methods: Four focus groups (FGs) were conducted with 21 physicians (13 residents and 8 attending physicians); two FGs in a university hospital and two in a regional hospital in French-speaking Switzerland. Discussions were audio-recorded and transcribed verbatim. Transcripts were analyzed thematically. Two of the four transcripts were double coded to establish coding framework consistency. Results: Participants reported that predictive tools were not routinely used after severe stroke, although most knew about such scores. Scores were reported as being useful in quantifying prognosis, advancing scientific evidence, and minimizing potential biases in decisions. Their use is, however, limited by the following barriers: perception of inaccuracy, general disbelief in scoring, fear of self-fulfilling prophecy, and preference for clinical judgement. Emotional and cognitive biases were common. Emotional biases distort clinicians’ knowledge and are notably: bias of personal values, negative experience, and cultural bias. Cognitive biases, such as availability, confirmation, and anchoring biases, that produce systematic deviations from rational thinking, were also identified. Conclusions: The results highlight opportunities to improve decision making in severe stroke through the promotion of predictive tools, strategies for communicating prognostic uncertainty, and minimizing cognitive biases among clinicians, in order to promote goal-concordant care.

## 1. Introduction

Stroke is the second leading cause of mortality and the third leading cause of disability worldwide [1]. Treatment decisions in acute stroke are complicated by prognostic uncertainty, especially regarding decisions about withholding or withdrawing life-sustaining treatment (LST). Clinicians and families face the challenge of providing goal-concordant care that aims to ensure consistency between the patients’ care preferences and the care provided [2]. During the first days after a severe stroke, patients often need LST, such as thrombolysis, hemicraniectomy, mechanical ventilation, antibiotics, and artificial nutrition or hydration [3]. While this acute period is essential for defining goals of care, it is fraught with multiple uncertainties regarding prognosis, the patients’ presumed preferences, and their anticipated ability to adapt to a new ‘normal’ after rehabilitation [4]. 

Goals of care conversations are also challenging because clinicians and families often have different ideas about prognosis [5], with a recent study describing prognostic discordance in over 50% of critically ill situations [6]. Families tend to be more optimistic and question clinicians’ plans for care [7]. Deficits and gaps in communication are a source of burden for all involved [8]. When patients lack decision-making capacity, their surrogate decision makers are called upon to make decisions on their behalves. While most patients prefer their family members to act as surrogate decision makers, [9], family members often feel unprepared for making LST decisions in acute situations given the uncertainty and the moral burden of such decisions [10,11,12]. Several strategies have been suggested to better support families facing LST decisions, but the effect of these interventions on psychological distress has been limited [13,14,15,16].

Decisions to continue LST are often grounded in hope of regaining an acceptable future state of life for patients [17]. Predictors of quality of life in stroke patients include stroke severity and neurological deficits, but also the patients’ ability to adapt to different life circumstances [18,19]. In order to improve the accuracy of predicting functional outcomes, several prognostic scores have recently been developed [20,21]. The Acute Stroke Registry and Analysis of Lausanne (ASTRAL) score is a validated prognostic instrument based on clinical criteria in acute settings [22]. Despite evidence indicating better predictive accuracy than clinical judgment alone, its implementation in routine practice remains uncommon [23,24]. 

Making decisions in the context of prognostic uncertainty is associated with clinically relevant cognitive biases [25]. These systematic deviations of rational thinking occur particularly when judgments are made intuitively and not based on reasoned deliberation [26,27,28]. Cognitive biases, such as anchoring bias, availability bias, or framing effect, have been shown to lead to erroneous medical treatment decisions [29]. In decision making about acute LST, biases may be associated with over- or under-estimated prognoses based on physicians’ previous personal experiences [30]. Erroneous prognostic estimates can lead to flawed decisions on LST, which are often irreversible [31]. 

In light of these challenges, the aims of this study were to explore (1) perspectives and experiences of clinicians regarding the use of predictive scores for LST decision making in severe acute stroke, and (2) clinicians’ awareness of their own cognitive biases in this context.

## 2. Methods

In order to capture personal, subjective approaches elicited by intersubjective discussion among colleagues, we conducted semi-structured focus group (FG) discussions with physicians in stroke units, ranging from residents in training to experienced attending physicians [32,33]. Physicians from two neurology services, one in a large university hospital and the other in a regional hospital, both based in the French-speaking part of Switzerland, were invited to participate via the heads of these services. Data were collected between April and June 2021. According to the Swiss law on research with human subjects, the research protocol was not subjected to review by the local ethics committee. 

We conducted four FG discussions, two in a university (UH) and two in a regional (RH) hospital. For each hospital, we organized two separate FGs, one for resident physicians and another for attending physicians, in order to promote free expression and avoid dominance or pressure due to hierarchy. We sought a diverse sample with regard to age and gender. The interview guide was established iteratively through consultation with experts in stroke neurology, neuro-palliative care, and neuroethics. The guide explored the clinicians’ perspectives regarding prognostic uncertainty, predictive scores, and LST decision making. Each of these major topics also included prompts to gain greater insights into subthemes such as scoring tools, cognitive biases, goal-concordant care, affective forecasting, changing goals of care, and multidisciplinarity. We focused our analysis on the scoring tools and biases in order to provide new insights. The FG discussions were moderated by experienced clinicians in neurology or palliative care (R.J.J., R.R.V.), accompanied by an expert in qualitative research (L.J.) and a final-year medical student (L.T.). FGs lasted between 40–55 min, were audio-recorded, anonymized, and transcribed verbatim. After four FGs, data saturation was discussed (L.T., L.J., R.R.V.), and as the semantic themes emerging from the FGs were relatively consistent, no additional FGs were conducted.

Thematic analysis was used to interpret the transcripts, which were structured according to our research questions, but new themes were coded inductively in order to allow for unanticipated themes and insights to be identified during analysis [34,35]. Firstly, three coders conducted an initial individual reading of the data set as a whole. Coders noted aspects of the data which responded to the research questions and drafted initial themes and codes. The three coders then discussed these themes and codes and developed a coding framework. Two of the four FGs were then coded in parallel by two researchers each (L.T., R.R.V. and L.T., L.J.), according to the previously developed framework. The coding for each of the FGs was compared, discrepancies were discussed, and changes to the framework were made. The final two FGs were coded by L.T. according to the revised framework, in consultation with the other two coders. As the FGs were conducted in French, translations were double checked by two native English-speakers.

## 3. Results

This qualitative study included 21 physicians (13 residents (code “Res”) and 8 attending physicians (code “Att”)) in 4 FGs (FG1–4) working in neurology and neuro-rehabilitation services (Table 1). 

### 3.1. Theme 1: Predictive Scores

Participants reported that prognostic scores were not commonly used in acute stroke neurology, and only one reported using the ASTRAL score in clinical practice, although most of the participants knew of it. We prompted physicians to discuss this in greater depth in order to understand the potential value and the barriers to the use of prognostic scores.

### 3.2. Potential Values of Prognostic Scores

Regarding the potential value of prognostic stroke scores (Table 2), some clinicians mentioned that scores could improve prognostic prediction and help explain the prognosis to the family by having quantitative scores. Others emphasized that these instruments allow for more rigorous measurements in clinical studies and advance science that can then promote evidence-based medicine. Finally, one clinician expressed the view that prognostic scores may reduce the impact of cognitive biases (which he expressed spontaneously before the topic of cognitive biases was introduced as another topic for the FG discussion).

### 3.3. Barriers to the Use of Prognostic Scores

The clinicians expressed several barriers to the use of prognostic stroke scores in practice (Table 3). Firstly, there was a fear that the scores may not be sufficiently reliable, especially regarding some clinical situations related to cognitive motor dissociation. Another argument against the use of these scores was a lack of trust in the validity of the score and the general relevance of scoring in clinical practice. One participant also raised concerns about a potential self-fulfilling prophecy: the scores may be overly pessimistic, then, because clinicians withdraw treatment as a result of predictive scores, thus reducing life expectancy in the stroke population and producing the bleak outcome they feared. Moreover, many clinicians reported that their hesitancy to use predictive scores was reinforced by the tendency of relying on one’s own clinical experience, and the fear of a choice dictated by statistics rather than the individual patients themselves. 

### 3.4. Theme 2: Emotional and Cognitive Biases

Some biases were brought up spontaneously, while others were discussed consciously after the moderator asked a question about their presence in clinical practice. We could identify in the FG discussions emotional biases, when emotional factors cause a distortion of knowledge depending on the clinician, and cognitive biases, defined as systematic deviations from rational thinking, not influenced by personal emotions. 

### 3.5. Emotional Biases

Three different types of emotional biases were identified inductively (Table 4). Bias of personal values were defined as instances when clinicians anticipate a patient’s or family’s behavior based on their own values. Bias of negative experience were identified when the clinician’s recollection of negative events in their personal or professional lives impacts their decisions. These negative experiences could be poor clinical outcome of stroke, but also other people’s negative reactions to a decision that the clinician had taken in the past. Finally, cultural bias occurs when clinicians have to adapt to a new cultural environment or face patients and families from a different cultural background but judge the situation exclusively from their own cultural reference system. 

### 3.6. Cognitive Biases

We also identified three common cognitive biases (Table 5). The availability bias refers to relying on the salient information immediately retrievable from memory, because it is common, serious, or recent. The confirmation bias happens when clinicians look for clues to support a rapidly and intuitively developed judgment (e.g., diagnosis, prognosis), while discarding information and values that conflict with this established judgment. The anchoring bias refers to relying on the information received first and the difficulty getting rid of it.

## 4. Discussion

Prognostication of the patient’s level of recovery is crucial for acute treatment decision making after severe stroke. Even though predictive scores have been developed and validated, their use in routine neurological practice is uncommon [11]. To our knowledge, the present study is the first to explore clinician’s perspectives of prognostic stroke scores and the impact of cognitive biases on acute LST decision making after severe stroke. 

Facing prognostic uncertainty, clinicians highlighted the value of being able to quantify prognosis by the help of a score, advancing scientific evidence for future practice, and limiting potential biases in treatment decisions. However, only one of the twenty-one clinicians who participated in our study actually uses the available scores. Multiple barriers to their use were identified: perceived inaccuracy, opposition or disbelief regarding scoring, the fear of self-fulfilling prophecy, attachment to clinical judgement, and feeling of a choice based on statistics rather than “the patients themselves”. Although no score will replace the clinician’s expertise and clinical assessment for a given patient, predictive scores could help navigate this uncertainty. Clinicians’ reluctance to use scores has already been described [12,36,37]; nevertheless, many prognostic scores have been shown to be more accurate than clinical judgement, even for experienced clinicians [9]. Clinicians’ estimate of terminally ill patients’ survival is also known to be inaccurate and most often overestimated, which impacts LST decisions [38]. Thus, reluctance to use predictive scores persists despite evidence of their validity. This discordance warrants future in-depth research, especially in an era where artificial intelligence is becoming more influential in decision making and health care systems more broadly [39]. 

Prognostication avoidance is notably related to high levels of uncertainty, projection of emotional consequences, as well as patient, family, and clinician’s anxiety about death [40]. Uncertainty may prompt clinicians to refrain from eliciting a prognosis as it could lead to early LST decision making. Uncertain prognoses could be used as an ethical justification for postponing LST decisions if it can be assumed that the prognostic accuracy will improve over time and goals of care are concordant with the patient’s wishes. In some cases, however, a family could be certain that the patient would not take the risk of an unacceptable outcome and refuse LST, even though the prognosis remains uncertain. Some also fear missing the opportunity to let the patient die when LST measures are still needed [41]. Although other opportunities to reconsider treatment decision are likely to arise in the patient trajectory of care, such reorientations may prove to be challenging for family and clinicians, and may require the support of palliative care specialists with experience in neuropalliative care [38].

As we have identified, clinicians fear withdrawing LST in situations where patients may eventually adapt to their new state of health better that they initially imagined. This corroborates previous research suggesting that families often describe uncertainty regarding the patient’s presumed wishes or their capacities to adapt to a new normal [42]. In fact, 87% of families wish to explicitly discuss these uncertainties and consider it both unavoidable and acceptable [43]. For them, talking about uncertainties is an essential part of the conversation leading to realistic expectations and genuine trust in clinicians. As clinician’s lead conversations about best medical options and prognosis, the family is invited to share the patient’s narrative and preferences. Since the disagreement between clinicians and families is widened by mutual misunderstanding of prognostic assessment [44], validated scores could be helpful in reducing this gap and facilitating these conversations. More specifically, training programs focused on prognostic assessment and predictive scores could facilitate their application in clinical practice.

Our study suggests that emotional and cognitive biases are common among clinicians working with acute stroke patients. It highlights the potential value of validated prognostic tools after severe stroke, as they could help reduce biases by providing more objective measures. These findings are similar to those in other fields such as psychology [45] or internal medicine [27]. The degree of introspection regarding the impact of biases varied; while some clinicians were not aware of them, some specifically acknowledged them. This begs the question of how clinicians could minimize the impact of such biases in their medical decision making. 

This study has several limitations. First, the recruitment was performed via the heads of the services, leading to a potential selection bias of the participating sample. Second, the ASTRAL score has been developed and validated in the university hospital where two FGs were performed, which could have influenced their views on predictive scores in stroke. Third, we only interviewed physicians, even though decision-making is an inter-professional endeavor in which nurses, therapists, psychologists, social workers, chaplains and others contribute. In addition, all FGs were conducted in French-speaking Switzerland; in order to reduce cultural biases specific to this region, similar research is needed in other regions. 

As uncertainty is inherent to medical decision-making, clinicians are at risk of systematic bias. Four strategies could decrease the impact of bias: health professional education, development of formal and explicit deliberation for decisional processes, application of operational prognostic tools and alternative assessment through substantive experts [26,31]. Further research is needed to develop such educational programs, deliberative processes and decisional tools adapted to the various clinical settings in which these decisions take place.

## Figures and Tables

**Table 1 brainsci-12-01591-t001:** Characteristics of study participants.

	UH (N = 12)	RH (N = 9)	Total (N = 21)
Hospital rank			
Resident	N = 8 (66%)	N = 4 (44%)	N = 12 (57%)
Attending	N = 4 (33%)	N = 5 (56%)	N = 9 (43%)
Field			
Neurology	N = 10 (83%)	N = 9 (100%)	N = 19 (90%)
Neurorehabilitation	N = 2 (17%)	N = 0 (0%)	N = 2 (10%)
Gender			
Female	N = 6 (50%)	N = 4 (44%)	N = 10 (48%)
Male	N = 6 (50%)	N = 5 (56%)	N = 11 (52%)

**Table 2 brainsci-12-01591-t002:** Potential value of prognostic scores.

Quantifying a prediction	“It can be helpful in discussions with the family (…) to quantify; We often do not have numbers and it can be useful to give them a general idea.” (Res5, FG1)
	“I think that, at a certain point, a score can improve the prediction because a score will eventually, at a certain moment, divide up a case.” (Att18, FG4)
Advancing science	“This is also what allows us to advance in medicine (…) you absolutely need scores for the trials, to assess your principal outcome measure, and then yes, we need scores to modernize medicine, to advance and find new therapies.” (Res14, FG3)
Limiting the impact of biases	“The scales, the scores, play a role in helping you orient yourself without the biases, especially the emotional biases, the psychological biases.” (Res15, FG3)

**Table 3 brainsci-12-01591-t003:** Barriers to the use of prognostic scores.

Barriers to the Use of Prognostic Scores	
Inaccuracy	“Certain patients do not show any initiative, who score very low on the scales but where there is a lot of cognitive function behind, unfortunately without being able to interact.” (Res9, FG2)
Opposition and disbelief	“To replace the fundamentals of medicine by scores, I am completely opposed to this, I say no.” (Res14, FG3)
	“There are many things to consider, I hardly see how that could be put into one single score.” (Res15, FG3)
Choice based on statistics rather than patients	“We will try to adapt the management of each patient, including the decision whether we take a palliative approach or do the maximum. It will be first and foremost a reflection and not a merely statistics.” (Res3, FG1)
	“You can have patients with the same score and completely different qualities of life.” (Res14, FG3)
Self-fulfilling prophecy	“If we have a score where those who apply their notions and those who say the score is valid as it is, are the same, then this makes a self-fulfilling prophecy.” (Att17, FG4)
Attachment to clinical experience	“I think that these scores are maybe less powerful insofar as it is actually less practical, and then in clinical practice it is true that we rely a lot on experience.” (Res1, FG1)“We do have guidelines, but we can also be artists along the way, I would say.” (Att19, FG4)

**Table 4 brainsci-12-01591-t004:** Emotional biases.

Emotional Biases	
Personal values	“I think there is a very bad conception, especially on the part of young physicians, because they think that they or someone from their family is like that, they think that the life of this or that should not be lived, even though they haven’t asked the patient.” (Att18, FG4)
	“Of course we have our own experience, which is subjective, of course we have a certain emotional aspect, “oh the patient looks like my father who died the year before”, this plays a big part.” (Att12, FG2)
Negative experiences	“I have become more careful, even if, now, this is against my conviction, I rather let the patient live longer if there is a possible pain that I feel in the entourage.” (Att11, FG2)
	“Aphasia, this is a bit the thing that I don’t like, the patient who arrives without information, the family that is not aware and the patient is aphasic, for me, that’s a bit the thing that I detest.” (Res14, FG3)
Cultural values	“In this case I adapt to the different values of people. Practically said, I go with the flow. If, for example, the nurses tell me that I exaggerate, that we should let the patient die, this tells me that the culture in the Vaud region is like that, that society is ready to let them die, so this is the value of the society.” (Att11, FG2)

**Table 5 brainsci-12-01591-t005:** Cognitive biases.

Cognitive Biases	
Availability	“We are all affected by a strange story (…) and then, at that moment, we pay close attention to the things we relate to these stories, of course we will search for things which are not always legitimate.” (Res7, FG1)
	“When he told me about his general context I completely switched and went with the idea of a cancer-associated myositis, all of this because I read about myopathies 3 months ago, an article which came out in Neurology.” (Res14, FG3)
Confirmation	“When we are tired, when we are in other situations, we fall a bit into these shortcuts due to previous experiences.” (Res7, FG1)
	“Whether it worked out well or not, we tend to reproduce. If it worked well, (we tend) to do it the same, and if it did not work well to avoid it and choose another path. I believe that it is instinctive and natural.” (Res15, FG3)
Anchoring	“I am sometimes amazed by the at times favorable course when I see them for follow-up. With cases where I had a preconceived idea in my head regarding the recovery.” (Att19, FG4)
	“When someone has acquired an awareness of something, it is very difficult to change it.” (Att9, FG2)

## Data Availability

The data presented in this study are available on request from the corresponding author.

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
