# Peer review of "‘We Have Guidelines, but We Can Also Be Artists’: Neurologists Discuss Prognostic Uncertainty, Cognitive Biases, and Scoring Tools"

_brainsci, 2022, doi:10.3390/brainsci12111591_

Round 1

Reviewer 1 Report

Dear Editor,

I reviewed the manuscript detailed below.

‘We have guidelines, but we can also be artists’: Neurologists discuss prognostic uncertainty, cognitive biases and scoring tools

The authors investigated in two hospitals how neurologists (n=21) deal with the situation after treating stroke patients with a severe stroke. I liked to read the paper, however the way the research was conducted and assembled is quite uncommon. The authors conducted semi-structured focus group discussions and interpreted statements provided by the participants. They especially focused on the knowledge regarding the predictive scores (e.g. ASTRAL score) and parameters characterizing the emotional bias. As mentioned above, it is interesting to read this paper, but it is not an original contribution in the genuine scope. I am uncertain if this kind of narrative science without interpreting quantitative results is suitable for the journal; the editor should judge this.      

Anyway, in my opinion this kind of paper should be published, the messages/ discussions are important and insights may help others in similar situations.

In this context I have some points which could improve the quality of the manuscript at some degree:

1.       Line 57-58: the authors indicate decisions made by physicians targets acceptable future state of life for patients and their families. I would temper the part with their families; we do medicine first of all for the patients and not for their families.  

2.       I would outline which scores they considered when the interviews were conducted.      

3.       There is a kind of polemic when stating phrases like “We are not like cardiologists with guidelines for all problems.” I would try to avoid things like that. Better, stay sober.

4.       I miss in the discussion the aspect that physician’s experience is important and no guideline or exchange with families etc. would replace his estimation for the individual case. I would give some comments on that.    

5.       In the interaction with families, I would suggest discussing, that physicians should incorporate an active role; physicians are supposed to know what the best option for patient’s current situation is; accordantly the discussions with the families should be guided. I would give some comments on that in the discussion.       

Author Response

Review Report Form 1

Thank you for all the valuable comments regarding our manuscript. We acknowledge that the value of qualitative research is to gain insight into the unique and subjective experience of the participants. As the first study to explore the view of clinicians on prognostic stroke scores and impact of cognitive biases after severe stroke, we believe this manuscript not only provides valuable insights into this topic, but that it also provides a starting point for further quantitative research into such biases.

Point 1. Line 57-58: the authors indicate decisions made by physicians targets acceptable future state of life for patients and their families. I would temper the part with their families; we do medicine first of all for the patients and not for their families.  

Response 1. Line 58: We focused on the patient’s future self and deleted “and their families” as suggested.

Point 2. I would outline which scores they considered when the interviews were conducted.      

Response 2. Most physicians considered the ASTRAL score, as added in the manuscript (line 123).

Point 3. There is a kind of polemic when stating phrases like “We are not like cardiologists with guidelines for all problems.” I would try to avoid things like that. Better, stay sober.

Response 3. Following your remark, the sentence has been removed from the paper.

Point 4. I miss in the discussion the aspect that physician’s experience is important and no guideline or exchange with families etc. would replace his estimation for the individual case. I would give some comments on that.    

Response 4. We acknowledged the value of clinicians’ expertise and assessment in the discussion (line 194-6).

Point 5. In the interaction with families, I would suggest discussing, that physicians should incorporate an active role; physicians are supposed to know what the best option for patient’s current situation is; accordantly the discussions with the families should be guided. I would give some comments on that in the discussion.       

Response 5. The role of physician in the goals-of-care conversation has been clarified as suggested (line 223-7), thank you for this comment. 

Reviewer 2 Report

The authors have created a work with a fascinating and sensitive topic! Unfortunately, many of us are aware of these gaps, but they must appear officially and especially through well-documented and carried-out research work.

However, I have a few remarks to make:

The authors could add the main ideas from the focus group discussion to the methodology and, based on which fundamental documentation, they chose those topics.

Also, a more precise description of the research methodology is mandatory so that it is reproducible (in stages).

What evaluation methods (specifically) did they refer to when they discussed the predictive scores (for example, the FIM proved to be a correct tool in predicting the functional capacity of the post-stroke patient, performed in the first 5-7 days after the stroke).

During the discussions, I would suggest that the authors also address the issue of training doctors and other health professionals on the correct evaluation methods, especially in neurology, where a multitude of evaluation scales have proven to have many psychometric qualities to be reliable assessment tools.

Author Response

Review Report Form 2

Thank you for your review and all the valuable comments regarding our manuscript, you will find our response below. 

Point 1. The authors could add the main ideas from the focus group discussion to the methodology and, based on which fundamental documentation, they chose those topics.

Response 1. The focus group discussion was based on three specific steps of acute management after severe stroke: prognosis, uncertainty and decision-making. From that point of view, we discussed several subthemes such as scoring tools, cognitive biases, goal-concordant care, affective forecasting, changing goals of care, and multi-disciplinarity. For the relevance of the manuscript, we focused our results on the scoring tools and biases as those data appeared to generate new and interesting hypotheses for further research. Further precisions on the methodology have been added in lines 95-100.

Point 2. Also, a more precise description of the research methodology is mandatory so that it is reproducible (in stages).

Response 2. The description of the methodology has been revised to add sufficient detail allowing replication of the analyses in future studies (lines 112-119). Please do not hesitate to inform us if further details about specific aspects are necessary.

Point 3. What evaluation methods (specifically) did they refer to when they discussed the predictive scores (for example, the FIM proved to be a correct tool in predicting the functional capacity of the post-stroke patient, performed in the first 5-7 days after the stroke).

Response 3. As only one attending physician attested using the ASTRAL score, we unfortunately do not have any other data on their evaluation methods. It would have been interesting indeed.

Point 4. During the discussions, I would suggest that the authors also address the issue of training doctors and other health professionals on the correct evaluation methods, especially in neurology, where a multitude of evaluation scales have proven to have many psychometric qualities to be reliable assessment tools.

Response 4. We thank you for this point and have added a sentence in lines 226-8.

Round 2

Reviewer 2 Report

The authors performed the requested adjustments.